

# Simulation of O₃ and NOₓ in Sao Paulo street urban canyons with VEIN (v0.2.2) and MUNICH (v1.0)

Mario E. Gavidia-Calderón[1], Sergio Ibarra-Espinosa[1], Youngseob Kim[2], Yang Zhang[3], and Maria de Fatima Andrade[1]

[1]Institute of Astronomy, Geophysics and Atmospheric Sciences, University of Sao Paulo, Sao Paulo, 05508-090, Brazil
[2]CEREA, Joint Laboratory École des Ponts ParisTech/EDF R&D, Université Paris-Est, 77455 Champs-sur-Marne, France
[3]Department of Civil and Environmental Engieneering, Northeastern University, Boston, MA 02115, USA

*Correspondence to*: Mario E. Gavidia-Calderón (mario.calderon@iag.usp.br)

**Abstract.** We evaluate the performance of the Model of Urban Network of Intersecting Canyons and Highways (MUNICH)
in simulating Ozone ($O_3$) and Nitrogen Oxides ($NO_x$) concentrations within the urban street canyons in the Sao Paulo Metropolitan Area (SPMA). The MUNICH simulations are performed inside Pinheiros neighborhood (a residential area) and Paulista Avenue (an economic hub), which are representative urban canyons in the SPMA. Both zones have air quality stations maintained by the Sao Paulo Environmental Agency (CETESB), providing data (both pollutants concentrations and meteorological) for model evaluation. Meteorological inputs for MUNICH are produced by a simulation with the Weather
Research and Forecasting model (WRF) over triple-nested domains with the innermost domain centered over the SPMA at a spatial grid resolution of 1 km. Street links coordinates and emission flux rates are retrieved from the Vehicular Emission Inventory (VEIN) emission model, representing the real fleet of the region. The VEIN model has an advantage to spatially represent emissions and present compatibility with MUNICH. Building height is estimated from the World Urban Database and Access Portal Tools (WUDAPT) Local Climate Zone map for SPMA. Background concentrations are obtained from the
Ibirapuera air quality station located in an urban park. Finally, volatile organic compounds (VOCs) speciation is approximated using information from Sao Paulo air quality forecast emission file and non-methane hydrocarbons concentration measurements. Results show an overprediction of $O_3$ concentrations in both study cases. $NO_x$ concentrations are underpredicted in Pinheiros but are better simulated in Paulista Avenue. Compared to $O_3$, $NO_2$ is better simulated in both urban zones. The $O_3$ prediction is highly dependent on the background concentration, which is the main cause for the model
$O_3$ overprediction. The MUNICH simulations satisfy the performance criteria when emissions are calibrated. The results show the great potential of MUNICH to represent the concentrations of pollutants emitted by the fleet close to the streets. The street-scale air pollutant predictions make it possible in the future to evaluate the impacts on public health due to human exposure to primary exhaust gases pollutants emitted by the vehicles.

## 1 Introduction

Street urban canyons are structures formed by a street and its flanked buildings (Oke et al., 2017). Due to their proximity to emissions from vehicles and their sides function as a compartment that limits pollutant dispersion, the street and the associated urban canyons are considered pollutant hotspots (Zhong et al., 2016). As more people start to live in urban areas (United Nations, 2018), and the ubiquity of urban canyons in the cities, pedestrians, commuters, bikers, and drivers are being

exposed to high pollutant concentrations every day (Vardoulakis et al., 2003). Consequently, the study of air pollution inside urban canyons is an important matter when dealing with studies of human health exposure related to traffic emissions.

To estimate the real impact of the pollutants on human health, it is necessary to obtain accurate pollutant concentrations and the lengths of exposure. Most cities are not covered by a high-density network of air quality stations. Even though the

measurements provide precise information, it is expensive and also very difficult to cover all of the impacted areas of a city (Zhong et al., 2016). One alternative, that is starting to be contemplated, is the use of numerical modeling to represent the pollutant behavior in urban canyons, which has the advantage of producing pollutant concentration information at high temporal and spatial resolutions.

Computational Fluid Dynamics (CFD) models are considered to be the best modeling approach to understand air pollutant dispersion inside the urban areas. Due to the limitations of high computational resources, these models cannot be applied for long time simulation periods nor for a large area (Fellini et al., 2019; Thouron et al., 2019).

A new type of model, the urban/local scale operational models, overcome these limitations by applying simplifications on

urban geometry and parameterizations of the mass transfer processes of air pollutants inside the urban canyons. The Operational Street Pollution Model (OSPM) and the Atmospheric Dispersion Model System (ADMS-urban) are two of the most popular operational models, which have already been tested for different cities around the world (Berkowicz et al., 1997; McHugh et al., 1997). Their main advantage is that they calculate pollutant concentrations when sources and receptors are in the same street urban canyon, but they present a limited treatment for the pollutant transfer between streets and

intersections (Carpentieri et al., 2012).

Street-network models are also operational, having the advantage of dealing with the transport of pollutants in city street intersections. The model SIRANE uses parametric relations to solve advection on street links, the dispersion in the street intersection, and interchange between the street and the over-roof atmosphere (Soulhac et al., 2011, 2012). Background

concentrations at the over-roof atmosphere are estimated using a Gaussian plume model. This estimation method inhibits a comprehensive atmospheric chemistry treatment.

Recently, the Model of Urban Network of intersecting Canyons and Highways (MUNICH) was developed by Kim et al. (2018) using a similar parameterization as SIRANE. MUNICH includes improvements in the treatment of the mean wind profile inside the urban canyon and the turbulent vertical mass transfer at the top of the street. It solves pollutant reactions using a chemical mechanism, so it can also simulate the production of ozone inside the urban canyons. MUNICH has been used to simulate ozone ($O_3$) and nitrogen oxides ($NO_x$) by Wu et al. (2020) in Tianhe District of Guangzhou city, and $NO_x$ as part of Street in Grid (SinG) model in Kim et al. (2018), Thouron et al. (2019) and Lugon et al. (2020) in the Paris region.

Significant information is required to run this kind of model. This is explained by Vardoulakis et al. (2003) that, in general, these models need at least information from traffic data, emissions, meteorological data, street geometry, and background concentrations. Recently, the VEIN model, a vehicular emission model, was developed by Ibarra-Espinosa et al. (2018) using information for Sao Paulo. VEIN is suitable to be used in street-network models because it uses the traffic flow, emission factors, and street morphology (i.e., intersection coordinates), to calculate the vehicular emissions. As a matter of fact, due to its architecture, it can be used together with MUNICH.

In Brazil, previous studies of air quality in urban canyons dealt with measurements of black carbon and $O_3$ inside a street canyon in Londrinas city center (Krecl et al., 2016), and dispersion of $NO_x$ was simulated in Curitiba with the ENVI-met model (Kruger et al., 2011). To our knowledge, this is the first study of modeling $O_3$ and $NO_x$ inside street urban canyons in Sao Paulo Metropolitan Area (SPMA) where it is very often the exceedance of $O_3$ state air quality standard (Andrade et al., 2017).

As the management of secondary pollutants remains a challenge in SPMA, the biggest megacity in South America, we aim to evaluate MUNICH operational street-network model to simulate $O_3$ and $NO_x$ concentration inside urban canyons, coupled with the VEIN emission model, to build a forecast system. This forecast system for air pollutant concentrations at street level can be used in air quality and traffic management of Sao Paulo neighborhood and in studies of health effects from traffic emission exposure.

## 2. Data and Methods

The experiment consisted of carrying out simulations of $O_3$, $NO_x$, NO, and $NO_2$ concentrations inside the SPMA urban street canyons with the MUNICH model. To evaluate model performance, the model results are compared against the measurements from Sao Paulo Environmental Agency (CETESB) air quality network. We choose Pinheiros urban area to test the model, where there is an air quality station in a mixed residential-commercial area. Once MUNICH and VEIN are calibrated, a study case is prepared by calculating the pollutant concentration inside Paulista Avenue, the economic central area of the city with high canyons. The selected study period covers the week from October 6[th] to October 13[th] of 2014. This



period is chosen before of no precipitation in SPMA, a period of high $O_3$ concentrations (Carvalho et al., 2015), the availability of data, and the availability of the emission inventory developed for a typical week in October 2014 (Ibarra-Espinosa et al., 2020).

## 2.1. MUNICH model

MUNICH is conceptually based on the SIRANE model (Soulhac et al., 2011). It has two main components, the street-canyon
component, which deals and solves pollutant concentrations inside the urban-canopy volume, and the intersection component, which calculates the pollutant concentrations inside the intersection volume. MUNICH differs from SIRANE in the treatment of the vertical flux by turbulent diffusion at the roof level (Schulte parameterization, Schulte et al., 2015) and in the mean wind velocity within the street canyon (Lemonsu parameterization, Lemonsu et al., 2004). Currently, MUNICH solves gas-phase pollutants based on the Carbon Bond mechanism version 5 (CB05). Further information is detailed in Kim
et al. (2018).

## 2.2 VEIN emission model

VEIN is an R package (R Core Team, 2019) to estimate vehicular emissions at a street level. VEIN imports functions from the package Spatial Features (Pebesma, 2018), which represent different types of geometries on space and perform geoprocessing tasks, from the data table package (Dowle and Srinivasan, 2020) to perform fast aggregation of databases, and
from the units package (Pebesma et al., 2016) to provide binding to the udunits library (https://www.unidata.ucar.edu/software/udunits/). VEIN includes a function to process vehicular flow at each street to generate activity traffic data, different emissions factors, and different sets of emissions calculation and post-processing tools (Ibarra-Espinosa et al., 2018). Specifically, the emissions factors are based on emissions certification tests with dynamometer measurements in laboratories (CETESB, 2015).

## 2.3 MUNICH input data

Urban canyon models required detailed input information, such as building height and street geometry. Their performance depends on the quality of this information (Vardoulakis et al., 2003). In recent years new tools have been developed to generate this information. Table 1 summarizes the model input used in this simulation experiment.




**Table 1. Summarized MUNICH input data.**

| Input data | Source |
| --- | --- |
| Meteorological input | WRF 3.7.1 simulation centered in SPMA (DX = 1km) |
| Street links coordinates and with lanes number | VEIN emission model (Ibarra-Espinosa et al., 2018) |
| Street links emissions | VEIN emission model (Ibarra-Espinosa et al., 2018) |
| Building height | World Urban Database and Access Portal Tools project (WUDAPT) database for SPMA (http://www.wudapt.org/) |
| Background concentration | $O_3$, NO, and $NO_2$ from the Ibirapuera Air Quality Station (AQS) |
| VOC speciation | Ethanol, Formaldehyde and acetaldehyde from WRF-Chem emission file from Andrade et al. (2015), other species are based from concentration showed in Dominutti et al. (2016) |

### 2.3.1 Emission and street links coordinates


Emission rates inside the street links in VEIN model are calculated using 104 million GPS vehicles coordinates in southeast Brazil (Ibarra-Espinosa et al., 2019). The GPS dataset is assigned to the OpenStreetMap (2017) dataset and once traffic flow is obtained, the vehicular compositions are generated and assigned with each emission factor reported by CETESB (2015). Emission factors are transformed into speed function, and then the average speed calculated at each street is used to obtain more representative emissions at each hour of a week. In addition, the estimation was calibrated with fuel consumption for




the year 2014. Ibarra-Espinosa et al. (2020) described all details regarding the emission estimation, with the emissions dataset in g h⁻¹ available at https://github.com/ibarraespinosa/ae1.

The emissions dataset presents two aspects that need to be discussed. The first one is that the regional emissions inventory,
which covered southeast Brazil, might not fully represent local reality. For instance, the ratio between travel demand models (TDM) and traffic flow of GPS for the study area is 2.22. Besides, the emission factors are average measurement of emissions certification tests, therefore, they may underestimate real-drive emissions (Ropkins et al., 2019). Furthermore, the real-world emission factors derived from tunnel measurements in São Paulo for $NO_X$ were 0.3 g km⁻¹ for light vehicles and 9.2 g km⁻¹ for heavy vehicles (Pérez-Martínez et al., 2014), while the respective fleet-weighted CETESB (2015) emission
factors are 0.44 g km⁻¹and 6.3 g km⁻¹, as shown on Fig. S1 in Supplement, resulting the ratios of 0.68 and 1.46. Therefore, if we consider the mean emission-factor ratio times the mentioned traffic flow ratio results that the $NO_X$ emissions should be approximately 2.37 higher.

Even when VEIN produces hourly emissions for a standard week, MUNICH only considers a standard day for weekdays and
weekends. We choose Wednesday emission as a typical weekday and Saturday emission for the weekend. Figure 1 shows the mean diurnal profile of $NO_x$ and VOCs emission fluxes from street-links in the Pinheiros neighborhood.

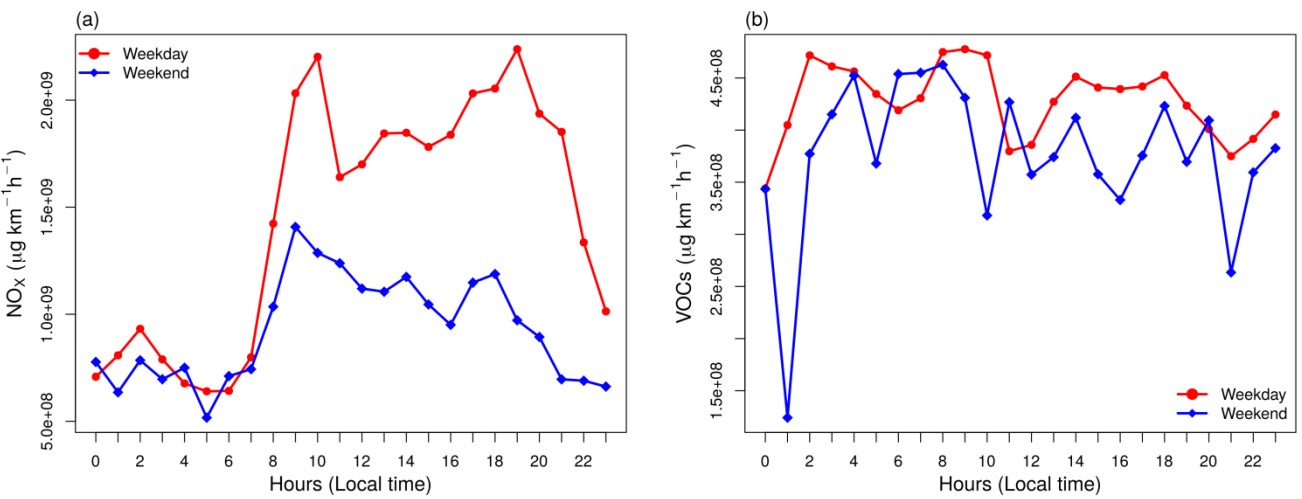

**Figure 1. Mean emission from all street links from the Pinheiros neighborhood for (a) $NO_X$ and (b) VOCs for typical weekday and**
**weekend.**



### 2.3.2 WRF simulation

Triple-nested domains are set up centered in SPMA. The mother domain has a spatial resolution of 25 km, the second 5 km, and the finest 1 km. The simulation at 1-km provides MUNICH with meteorological information. Initial and boundary conditions are retrieved from Historical Unidata Internet Data Distribution (IDD) Gridded Model Data
(https://rda.ucar.edu/datasets/ds335.0/index.html). Table 2 shows WRF configuration and Fig. 2, the WRF domains.

**Table 2. WRF simulation configuration.**

| Attribute | Configuration |
| --- | --- |
| WRF version | 3.7.1 |
| Domains spatial resolution | DX= 25 km, 5 km and 1 km |
| Simulation period | October 3$^{rd}$ to October 13$^{th}$, 2014 (three first days are spin-up days and not analyzed) |
| Meteorological IC/BC | Historical Unidata Internet Data Distribution (IDD) Gridded Model Data (DS0335) |
| Longwave Radiation | RRTMG (Iacono et al., 2008) |
| Shortwave Radiation | RRTMG (Iacono et al., 2008) |
| PBL | YSU (Hong et al., 2006) |
| Surface Layer | Noah (Tewari et al., 2004)) |
| Cumulus cloud | Multi-scale Krain-Fritsch (Zheng et al., 2016) |
| Cloud Microphysics | Morrison double-moment (Morrison et al., 2009) |


**Figure 2. WRF simulation domains for domains of 25 km (D01), of 9 km (D02) , and of 1 km (D03) spatial resolution. D03 provides the meteorological information to MUNICH, Sao Paulo city is outlined in thick black line and the red dots shows MUNICH domains location.**

Before using the WRF simulation outputs for MUNICH modeling, a model verification is performed. We use benchmarks

suggested by Emery et al. (2001), which were also used in Reboredo et al. (2015) and Pellegati et al. (2019). However, Monk et al. (2019) explained that these benchmarks are suitable for domains in "simple" terrain, they also presented other sets of benchmarks for "complex" terrain, the latter being more suitable for SPMA. The results are detailed in Table 3. The temperature at 2 m (T2) and relative humidity at 2 m (RH) reach the simple terrain benchmarks while wind speed and direction at 10 m (WS10 and WD10, respectively) are very close to them. When compared against complex terrain

benchmarks, only the mean bias of WD10 is beyond the benchmark. Finally, T2, RH, and WS10 satisfy the good performance criteria of Keyser and Anthes (1977) and Pielke (2013). More details are shown in Tables S1 and S2 in the Supplement.

**Table 3. WRF statistical model verification of simulation quality.**

| Parameter | Benchmark Simple terrain | Benchmark Complex terrain | Value from the WRF simulation |
|---|---|---|---|
| Temperature at 2m | MB[a] $< \pm 0.5$ K | MB $< \pm 1.0$ K | 0.27 K |
| | MAGE $< 2.0$ K | MAGE $< 3.0$ K | 1.59 K |
| | IOA $\geq 0.8$ | | 0.83 K |
| Relative humidity at 2m | MB $< \pm 10.0$ % | | -5.02 % |
| | MAGE $< 20$ % | | 9.79 % |
| | IOA $> 0.6$ | | 0.74 |
| Wind speed at 10 m | MB $< \pm 0.5$ m.s$^{-1}$ | MB $< \pm 1.5$ m.s$^{-1}$ | 0.79 m.s-1 |
| | RMSE $\leq 2$ m.s$^{-1}$ | RMSE $\leq 2.5$ m.s$^{-1}$ | 1.59 m s-1 |
| Wind direction at 10 m | MB $< \pm 10.0$ ° | MB $< \pm 10.0$ ° | **-16.23 °** |
| | MAGE $< 30$ ° | MAGE $< 55$ ° | 55 ° |

[a] MB: Mean bias, MAGE: Mean absolute gross error, IOA: Index of agreement and RMSE: Root mean square error. Results outside the benchmark are highlighted in bold.

### 2.3.3 Building height and street width

Building height is retrieved from the World Urban Database and Access Portal Tools project (WUDAPT) for SPMA (Fig. 3). WUDAPT classifies urban areas into 17 Local Climate Zones (LCZ). These LCZ are divided into build types, which are LCZ from 1 to 10, and land cover types, which go from A to G. Each of these LCZ presents different thermal, radiative, surface cover, and geometric properties. The building height is the height of roughness elements, which is the geometric average of building heights (Stewart and Oke, 2012). The WUDAPT file for SPMA is a raster with a spatial resolution of 120 m and was previously used in Pellegati et al. (2019).

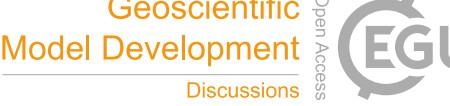

The number of lanes is provided by the OpenStreetMap dataset, so the street width is calculated by using 3 m of lane width and by adding 1.9 m to each side of the street as sidewalk width. Most OpenStreetMap streets do not include the number of

lanes for this region, therefore, they are hole-filled with the average by type of street.

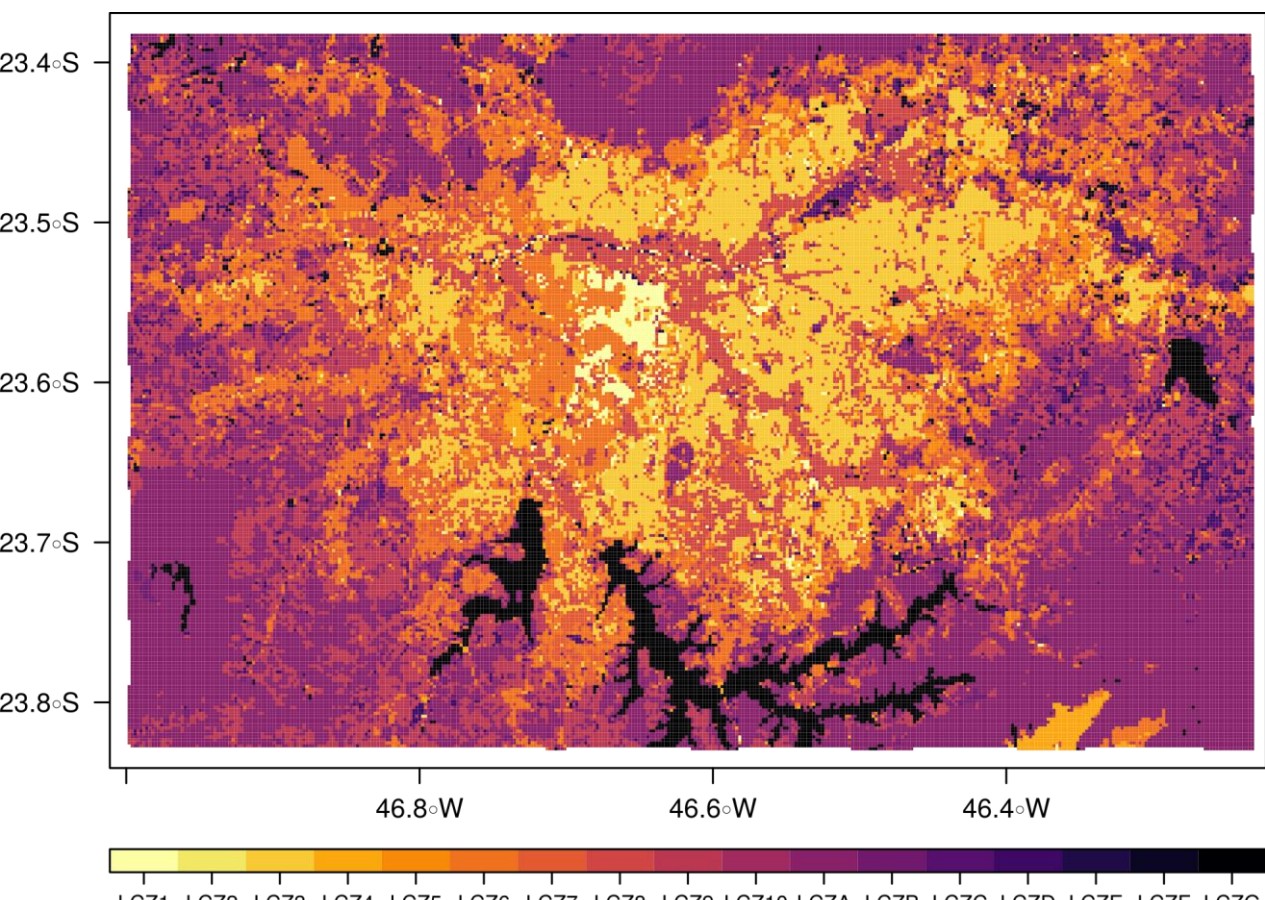

**Figure 3. Local Climate Zones for SPMA.**

**2.3.4 Background concentration**

Vardoulakis et al. (2003) explained that the background concentration in street modeling is necessary to include the proportion of air pollutants that are not emitted inside the street links. In the SinG model, background concentrations are the concentrations calculated by Polair3D, a mesoscale air quality model (Kim et al., 2018). Wu et al. (2020) chose as the background concentration, measurements from a station located very close to the study zone. Consequently, we consider

background concentration the concentration outside the MUNICH domain. With that in mind, by using the mean wind field from WRF simulation for the study period, we select Ibirapuera AQS (83 shown in Fig. 4) measurement as background





concentration, which, according to the wind field, advect pollutants to Pinheiros station (99) and Cerqueira Cesar (83) as can be seen in Fig. 4.


**Figure 4. WRF average wind field for the simulation period with CETESB air quality stations (AQS). The green diamond shows Pinheiros AQS (99), the red diamond shows Cerqueira Cesar AQS (83), and the blue diamond shows Ibirapuera AQS (83).**





## 2.4 Measurements and statistical analysis

Meteorological and air pollutant measurements are retrieved from CETESB air quality network. To evaluate WRF
simulation in the finest domains, observations from 41 air quality stations (AQS) are used. Background concentration comes
from Ibirapuera AQS.  Pinheiros AQS is used to evaluate MUNICH performance in the Pinheiros neighborhood, while
Cerqueira Cesar is used to evaluate Paulista Avenue. To evaluate model performance we follow the recommendations from
Emery et al. (2017). We also use the evaluation statistics from Hanna and Chang (2012): Fractional bias (FB), Normalized
mean-square error (NMSE), Fraction of predictions within a factor of two (FAC2), and normalized absolute difference
(NAD) . The acceptance criteria for urban zones are: |FB| <= 0.67, NMSE <= 6, FAC2 >= 0.3 and NAD <= 0.5.

## 2.5 Model set up

We use MUNICH to simulate two urban areas inside SPMA, the first domain is Pinheiros neighborhood and the second one
is Paulista Avenue. VEIN calculates the emissions for the whole SPMA, so we retrieve $NO_x$, $NO_2$, and VOCs emissions for
the streets located in both domains. In MUNICH, NO emissions are estimated from $NO_x$ and $NO_2$ emissions.

Figure 5 shows MUNICH domain for the Pinheiros neighborhood and Paulista Avenue. The yellow dot represents the
location of the air quality stations. The red lines are the street links used by VEIN to calculate the emissions, and the yellow
rectangle the urban canyon selected for comparison against observation.

There are 677 street links for Pinheiros and 535, for Paulista Avenue. Nine points of WRF simulation cover the Pinheiros
domains, while twelve WRF points represent Paulista Avenue domains. From WUDAPT we can see that inside Pinheiros
there is a variety of buildings with different heights. Pinheiros AQS is located in an urban canyon that has a mean building
height of 5 meters (LCZ 6 - Open Low Rise). On the other hand, Paulista Avenue domain is more uniform, presenting urban
canyons with a mean building height of 45 meters (LCZ1 - Compact high rise).

**Figure 5. Pinheiros neighborhood (a) adn Paulista Avenue (b) MUNICH domains and building height, the red lines are the streets considered in VEIN, the yellow dot shows Pinheiros AQS and Cerqueira Cesar (AQS). Yellow squares highlight the selected urban canyon for comparison against observation. At the bottom, satellite photos of those urban canyons (Source: © 2019 Google, Image © 2019 Maxar Technologies).**

## 3 Results

Here we present the $O_3$ and $NO_x$ simulations with MUNICH for a week of October 2014. We first calibrated the input emissions by studying Pinheiros neighborhood, to later simulate $NO_x$ inside Paulista Avenue urban canyon.

### 3.1 Control case for the Pinheiros neighborhood

Figure 6 shows the results of MUNICH simulation using the original emissions calculated by VEIN for SPMA. MUNICH simulations are very close to background concentrations, which leads to an overprediction of $O_3$ and underpredicted NO and $NO_X$ concentrations. This is produced by a dependence of MUNICH on background concentration and by emission underestimation. The emission underestimation is caused by emission factors calculated based on average measurements of emissions certification tests, and because emission factors derived from dynamometer, and cycle measurements do not



represent real-drive emissions (Ropkins et al., 2019). It's also probable that the number of vehicles could have been underestimated inside the urban canyon. The underestimation of $NO_X$ is caused by the underestimation of NO concentrations. $NO_2$ concentration magnitude is well represented by MUNICH.

245

The diurnal variation of MUNICH simulation, observation, and background concentrations are shown in Fig. 7. MUNICH simulated coherently the temporal variation of $O_3$ and $NO_2$ concentration inside the urban canyon. For NO and $NO_x$, the temporal variation during the day and until midnight is well simulated, while the morning peak at 6 hours is underestimated. After midnight, a higher concentration of $NO_X$ occurs by the increase of heavy-duty vehicles at night that mainly run with diesel. In Pinheiros urban canyons, there is predominantly a flow of light-duty vehicles, even though it is registered high $NO_X$ concentrations that it's transported from the highway. The mean difference between MUNICH simulation and background concentration for $O_3$, $NO_x$, NO, and $NO_2$ are -13.10 $\mu$g m$^{-3}$, 28.61 $\mu$g m$^{-3}$, 9.25 $\mu$g m$^{-3}$, and 14.43 $\mu$g m$^{-3}$, respectively.



255

**Figure 6. Comparison of MUNICH results against background and observation concentrations of (a) O$_3$, (b) NO$_X$, (c) NO, and (d) NO$_2$ for Pinheiros urban canyon from the control case.**



**Figure 7. Diurnal profile of MUNICH results, background, and concentrations of (a) O$_3$, (b) NO$_X$, (c) NO, and (d) NO$_2$ for Pinheiros urban canyon from the control case.**

### 3.2 Emission adjustment

We ran different scenarios with increased NO$_X$ and VOCs emission from VEIN. The best results were produced when doubled the NO$_X$ and VOCs emissions, this scenario is called MUNICH-Emiss. With this adjustment, we achieve an overall improvement of MUNICH simulations. Figure 8 shows the new comparison between the model, background concentration, and observations. O$_3$ is still overpredicted which is caused by the higher value of O$_3$ background concentration together with a low NO background concentration; nevertheless, the simulated O$_3$ concentration during night is well represented and daily peaks values are closer to observations.







**Figure 8. Comparison of MUNICH results against background and observation concentrations of (a) O₃, (b) NOₓ, (c) NO, and (d) NO₂ for Pinheiros urban canyon from the MUNICH-Emiss simulation.**

$NO_x$ and NO simulations are still underpredicted, but $NO_2$ is in the same magnitude as observations. $NO_x$ underprediction is still mainly attributed to the underprediction of NO, especially during October 8[th], 9[th] and 10[th] where high observational values of NO were recorded. However, MUNICH can better represent the observed high concentration during Saturday 11[th], as MUNICH uses the same emission profile for the weekend and weekdays, this high simulated NO concentration resulted from the influence of meteorology.

Figure 9 shows the diurnal profiles for this simulation. The new MUNICH-Emiss profiles are closer to observed concentration profiles, with a better representation of the peak concentrations magnitude of $NO_x$, NO, and $NO_2$. The mean difference over the simulation period between simulated and the background concentrations for O₃, $NO_x$, NO, and $NO_2$ are -17.85 µg m⁻³, -57.26 µg m⁻³, 23.60 µg m⁻³, and 21.07 µg m⁻³, respectively, showing bigger differences than the control case previous scenario and the influence of the reaction with NO emissions.





Figure 9. Diurnal profile of MUNICH results, background, and concentration for (a) O₃, (b) NOₓ, (c) NO, and (d) NO₂ for Pinheiros urban canyon from the MUNICH-Emiss simulation.

Table 4 summarizes the performance statistics for each scenario and background. The performance statistics from the MUNICH-Emiss case show lower values of MB, NMGE, and RMSE for all pollutants, except NO₂ that presents a slightly increase in these indicators. They also show high values of R ($\geq 0.7$) for each pollutant in every case, which indicates that the temporal variations of emission and background concentration are in the same phase as the observations. In general, in both MUNICH simulations, NO₂ and O₃ are better simulated. MUNICH-Emiss case performs better and also achieves the recommendations of Hanna and Chang (2012) for O₃, NO₂ NO, and NOₓ, whereas MUNICH control case didn't reach these recommendations for NO.





**Table 4. Statistical indicators for $O_3$, $NO_X$, NO, and $NO_2$ for comparison between background concentration, MUNICH simulation, and MUNICH-Emiss against observation from Pinheiros AQS.**

| | | $\bar{M}^b$ | $\bar{O}$ | $\sigma_M$ | $\sigma_O$ | MB | NMB | NMGE | RMSE | R | \|FB\| | NMSE | FAC2 | NAD |
|---|---|---|---|---|---|---|---|---|---|---|---|---|---|---|
| $O_3$ | Background | 67.6 | 41.5 | 63.2 | 47.5 | 26.1 | 0.6 | 0.6 | 32.4 | 1.0 | **0.5** | **0.4** | **0.5** | **0.2** |
| | MUNICH | 54.5 | 41.5 | 62.1 | 47.5 | 13.0 | 0.3 | 0.3 | 22.2 | 1.0 | **0.3** | **0.2** | **0.6** | **0.1** |
| | MUNICH-Emiss | 49.7 | 41.5 | 59.5 | 47.5 | 8.2 | 0.2 | 0.3 | 18.0 | 1.0 | **0.2** | **0.2** | **0.6** | **0.1** |
| $NO_X$ | Background | 60.3 | 146.4 | 37.3 | 150.3 | -86.0 | -0.6 | 0.6 | 149.6 | 0.8 | 0.8 | **2.5** | **0.5** | **0.4** |
| | MUNICH | 88.9 | 146.4 | 57.4 | 150.3 | -57.4 | -0.4 | 0.5 | 128.5 | 0.7 | **0.5** | **1.3** | **0.7** | **0.2** |
| | MUNICH-Emiss | 117.6 | 146.4 | 85.6 | 150.3 | -28.8 | -0.2 | 0.5 | 120.0 | 0.6 | **0.2** | **0.8** | **0.7** | **0.1** |
| NO | Background | 9.5 | 54.6 | 12.7 | 88.9 | -45.1 | -0.8 | 0.8 | 91.5 | 0.8 | 1.4 | 16.2 | **0.3** | 0.7 |
| | MUNICH | 18.7 | 54.6 | 28.7 | 88.9 | -35.9 | -0.7 | 0.8 | 80.7 | 0.7 | 1.0 | 6.4 | 0.1 | **0.5** |
| | MUNICH-Emiss | 33.1 | 54.6 | 48.5 | 88.9 | -21.5 | -0.4 | 0.8 | 74.5 | 0.6 | **0.5** | **3.1** | **0.3** | **0.2** |
| $NO_2$ | Background | 45.8 | 62.7 | 23.4 | 25.9 | -16.8 | -0.3 | 0.3 | 21.2 | 0.9 | **0.3** | **0.2** | **0.9** | **0.2** |
| | MUNICH | 60.3 | 62.7 | 22.8 | 25.9 | -2.4 | 0.0 | 0.2 | 13.3 | 0.9 | **0.0** | **0.0** | **1.0** | **0.0** |
| | MUNICH-Emiss | 66.9 | 62.7 | 22.0 | 25.9 | 4.2 | 0.10 | 0.2 | 14.8 | 0.8 | **0.1** | **0.1** | **0.9** | **0.0** |

[b] $\bar{M}$ - Model value mean ($\mu g\ m^{-3}$), $\bar{O}$ - Observation mean ($\mu g\ m^{-3}$), $\sigma_M$ - model standard deviation ($\mu g\ m^{-3}$), $\sigma_O$ - observation standard deviation ($\mu g\ m^{-3}$), MB - mean bias ($\mu g\ m^{-3}$), NMB - normalized mean bias, NMGE - normalized mean gross error, RMSE - root mean square error ($\mu g\ m^{-3}$), R - correlation coefficient, FB - fractional mean bias, NMSE - normalized mean-square error, FAC2 - fraction of predictions within a factor of two , and NAD - normalized absolute difference. Values in bold satisfied Hanna and Chang (2012) acceptance criteria.

300    Figure 10 shows the mean hourly concentration of $O_3$ and $NO_x$ in the Pinheiros neighborhood, the red diamond points to the location of Pinheiros air quality station. Because the VEIN model can distribute spatially the emissions, there is a variation of concentrations in different street links. For example, the orange diamond shows the location of a traffic light, where traffic jams occur, causing lower $O_3$ concentrations from higher $NO_X$ emissions.





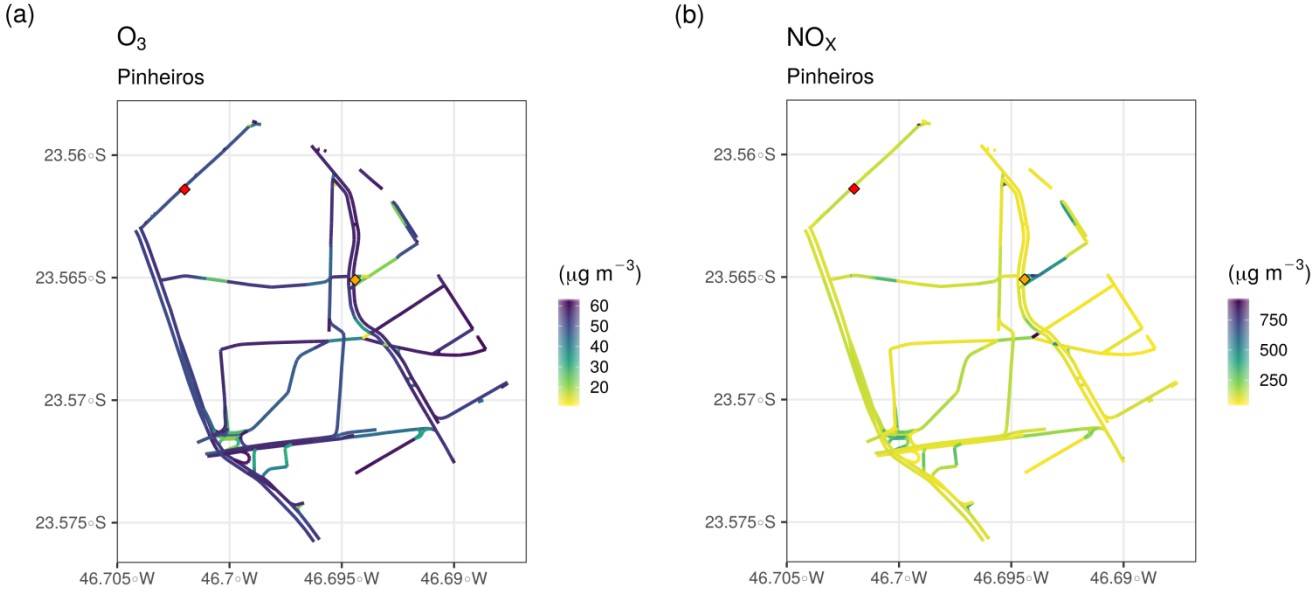

**Figure 10. Hourly mean simulated concentration of (a) $O_3$ and (b) $NO_X$ for Pinheiros neighborhood. Red diamond denotes the location of the Pinheiros AQS and orange diamond denotes traffic light location.**

We also perform additional sensitivity simulation by running MUNICH scenario using the background concentrations from Santos AQS (light blue diamond in Fig. 4). Compared to the Ibirapuera AQS site, measured $O_3$ and $NO_2$ concentrations are lower and those of NO concentrations are higher at the Santos AQS. This results in $O_3$ and $NO_2$ underprediction and a better simulation of NO concentration magnitude; however, all evaluated pollutants present lower R values and higher NMGE values than MUNICH-Emiss scenario with Ibirapuera AQS as background concentration. Simulated $NO_2$ and $O_3$ follow background concentrations, which indicates that the MUNICH simulations have a strong dependence on the background concentration (see Fig. S2 and S3 in Supplement).

Lastly, a sensitivity simulation was performed with an only increase of $NO_x$ emission by four and remaining VOCs original emission using Ibirapuera background concentration. This results in a better $O_3$ representation but unrealistic $NO_x$, NO, and $NO_2$ concentration (see Fig. S4 and S5 in Supplement). As SPMA has a COV-limited regime (Andrade et al., 2017), the increment of $NO_X$ emission with lead to a reduction of $O_3$ concentration. Many studies have shown that Sao Paulo atmosphere is VOC-limited (Schuch et al., 2020) due to the high $NO_X$ emission by the heavy-duty that are under old emissions regulations. The new regulations for diesel engine emissions was established recently and are being implemented according to the recycle of the fleet, that is 20 years of use for diesel trucks (CETESB, 2019).





### 3.3 Aplication for the Paulista Avenue

325    The MUNICH simulation is performed with calibrated emissions for a domain that contains a well-defined urban canyon, the Paulista Avenue. The simulation shows a better representation of $NO_x$, NO, and $NO_2$ temporal variation and a good representation of concentration magnitude (Fig. 11). Although the MB indicates an overprediction of $NO_x$, NO, and $NO_2$ (Table 5), Figure 12 shows that this is caused by an overprediction of these pollutants during night hours, linked to a mismatch of emissions. Statistics in Table 5 shows an improvement in representing concentration magnitudes of $NO_x$, NO,

330    and $NO_2$ with mean simulated concentrations close to observations and very low values of MB, NMB, and RMSE. In this case, R values are lower than those in the Pinheiros case but still higher than 0.5, confirming that there is a mismatch of simulated concentrations, which is clearer in MUNICH $NO_X$ and NO peak happening before observation. The MUNICH-Emiss simulations achieve Hanna and Chang (2012) performance criteria for $NO_x$ and $NO_2$. $NO_2$ is the best simulated species.

335



**Figure 11.** Comparison of MUNICH results against background and observation concentration for (a) $O_3$, (b) $NO_X$, (c) NO, and (d) $NO_2$ for Paulista Avenue urban canyon. Note that no $O_3$ observation for Paulista Avenue.



**Figure 12. Diurnal profile of MUNICH results, background, and concentration for (a) O$_3$, (b) NO$_X$, (c) NO, and (d) NO$_2$ for Paulista. Note that no O$_3$ observation for Paulista Avenue.**





**Table 5. Statistical indicators for O$_3$, NO$_X$, NO, and NO$_2$ for comparison between background concentration and MUNICH-Emiss against observation from Cerqueira Cesar AQS.**

|  |  | $\bar{M}^c$ | $\bar{O}$ | $\sigma_M$ | $\sigma_O$ | MB | NMB | NMGE | RMSE | R | \|FB\| | NMSE | FAC2 | NAD |
|---|---|---|---|---|---|---|---|---|---|---|---|---|---|---|
| NO$_X$ | Background | 56.8 | 105.8 | 36.6 | 66.8 | -49.0 | -0.5 | 0.5 | 68.9 | 0.7 | **0.6** | 0.8 | **0.6** | **0.3** |
|  | MUNICH-Emiss | 114.8 | 105.8 | 68.4 | 66.8 | 9.0 | 0.1 | 0.6 | 74.2 | 0.4 | **0.1** | **0.5** | 0.7 | **0.0** |
| NO | Background | 7.3 | 26.9 | 10.3 | 30.7 | -19.6 | -0.7 | 0.8 | 32.5 | 0.6 | 1.1 | **5.3** | 0.2 | 0.6 |
|  | MUNICH-Emiss | 28.0 | 26.9 | 35.2 | 30.7 | 1.1 | 0.0 | 1.1 | 40.8 | 0.2 | **0.0** | 2.2 | 0.2 | **0.0** |
| NO$_2$ | Background | 45.5 | 64.6 | 24.3 | 26.5 | -19.0 | -0.3 | 0.3 | 24.2 | 0.8 | **0.3** | **0.2** | **0.8** | **0.2** |
|  | MUNICH-Emiss | 71.9 | 64.6 | 23.9 | 26.5 | 7.4 | 0.10 | 0.2 | 19.1 | 0.8 | **0.1** | **0.1** | **0.9** | **0.1** |

$^c$ $\bar{M}$ - Model value mean (µg m$^{-3}$), $\bar{O}$ - Observation mean (µg m$^{-3}$), $\sigma_M$ - model standard deviation (µg m$^{-3}$), $\sigma_O$ - observation standard deviation (µg m$^{-3}$), MB - mean bias (µg m$^{-3}$), NMB - normalized mean bias, NMGE - normalized mean gross error, RMSE - root mean square error (µg m$^{-3}$), R - correlation coefficient, FB - fractional mean bias, NMSE - normalized mean-square error, FAC2 - fraction of predictions within a factor of two , and NAD - normalized absolute difference. Values in bold satisfied Hanna and Chang (2012) acceptance criteria.

## 4 Discussion and conclusions

Simulating air pollutants inside urban street canyons is a challenging task. It is even more difficult in cities as heterogeneous as Sao Paulo, where its urban structure is not always textbook defined. The limited number of air quality stations located inside or near urban canyons, together with the lack of information from detailed emission inventories and urban morphology data, hinder accurate air quality modeling, and consequently the air quality management.

In this paper, we attempt to fill in this gap by using the MUNICH street-network model together with the VEIN vehicular emissions model. The latter provides temporal and spatially detailed emission fluxes inside the main streets and coordinates and width of the streets (i.e., the street network). The urban morphology is completed by extracting the building height from the WUDAPT database for Sao Paulo Metropolitan Area. The advantages of using MUNICH are that, besides solving pollutant dispersion, it also solves photochemistry reactions and its nature of operational model to solve pollutant concentration at neighborhood scale considering also street intersections.

Results showed that MUNICH simulations that used adjusted emissions can better represent the temporal variation of O$_3$, NO$_x$, NO, and NO$_2$ concentrations inside urban canyon. Nevertheless, the results are highly dependent on background concentrations and emission fluxes. This background concentration dependence is stronger in secondary pollutants such as O$_3$, and primary pollutants are more determined by emission fluxes. The reason for the significant contribution of background concentration is that MUNICH is based in SIRANE, and SIRANE also presents a significant contribution from background concentration (Soulhac et al., 2012).




The main cause of $O_3$ overprediction in our simulation for both tested urban zones is the high value of background $O_3$ concentration measured in Ibirapuera AQS. In Pinheiros neighboorhood, the underprediction of NOX concentration is caused by the underprediction of NO concentration in Pinheiros during the second half of the week. The concentration magnitudes in Paulista Avenue are well represented but there was a mismatching with observed concentration. MUNICH with the adjusted emissions fulfills the performance criteria. $O_3$ concentration simulated in Pinheiros and Paulista Avenue is less than background concentrations, these same results are reported by Wu et al. (2019). As noted in Krecl et al. (2016), this behavior is caused by the high $NO_X$ emissions inside the street urban canyons, which rapidly deplete the formed $O_3$ and the one from the rooftop (i.e, background concentration).

As the main source of superficial NO and $NO_2$ emissions in São Paulo are vehicles, it is necessary to go deeper into the reasons why the scenario MUNICH-Emiss performs better. The increase of the emissions is necessary because the emissions factors are the average of emission certification tests (CETESB, 2015). It has been shown that emission factors derived from dynamometer and cycle measurements do not represent real-drive emissions (Ropkins et al., 2009). São Paulo does not have an Inspection and Maintenance (I&M) program, therefore, may exist a fraction of the fleet which are high emitters and do not meet the emission standards, more details can be found in Ibarra-Espinosa et al. (2020). Furthermore, the comparison of traffic flow between GPS and TDM data for Pinheiros area showed that TDM traffic flows are 2.22 times higher than GPS. Hence, more representative traffic flows would also improve the emissions compilation. As a conclusion, it is important to develop new and more representative vehicular traffic flow and emission factors for Brazil.

With calibrated emissions, the good performance of MUNICH in representing $NO_2$ concentrations in both neighborhoods and NO and $NO_x$ in Paulista Avenue urban canyon suggests that VEIN model distributes emissions spatially and temporally efficiently, which proves its potential to be used in other cities. VEIN is being continuously developed and currently offers some utilities to format emissions to the MUNICH model. On the other hand, now Google Earth allows new features as 3D view, where information on building height can be retrieved. These new features can be used to improve MUNICH input data, and therefore, the model simulation results. Further, a better estimation of background concentrations from photochemical grid models can potentially improve the model performance.

The results obtained show the promising capability of MUNICH to represent the concentrations of pollutants emitted by the fleet close to the streets. As MUNICH uses the CB05 gas-phase mechanism, it can also simulate VOCs inside the urban canyon. Measurements of VOCs inside urban canyons are therefore necessary to validate the model in the future. An accurate prediction of street-scale air pollutant concentrations will enable the future assessment of the impacts on human health due to their exposure to air pollutants emitted by the vehicles.






## Appendix A: Statistical indicators

**Table A6. Statistical indicator definition.**

| Statistical indicator | Definition | Reference |
|---|---|---|
| Fraction of prediction within a factor of two (FAC2) | $FAC2 = 0.5 \leq \dfrac{M_i}{O_i} \leq 2.0$ | Emery et al (2017) |
| Mean Bias (MB) | $MB = \dfrac{1}{N}\sum_{i=1}^{N}(M_i - O_i)$ | Emery et al. (2017) |
| Mean Absolute Gross Error (MAGE) | $MAGE = \dfrac{1}{N}\sum_{i=1}^{N}|M_i - O_i|$ | Emery et al. (2017) |
| Normalized mean bias (NMB) | $NMB = \dfrac{\sum_{i=1}^{N}(M_i - O_i)}{\sum_{i=1}^{N}O_i}$ | Emery et al. (2017) |
| Normalized mean error (NME) | $NME = \dfrac{\sum_{i=1}^{N}|M_i - O_i|}{\sum_{i=1}^{N}O_i}$ | Emery et al. (2017) |
| Root mean square error (RMSE) | $RMSE = \sqrt{\dfrac{1}{N}\sum_{i=1}^{N}(M_i - O_i)^2}$ | Emery et al. (2017) |
| Correlation coefficient (R) | $R = \dfrac{1}{(N-1)}\sum_{i=1}^{N}\left(\dfrac{M_i - \bar{M}}{\sigma_M}\right)\left(\dfrac{O_i - \bar{O}}{\sigma_O}\right)$ | Emery et al. (2017) |
| Fractional mean bias (FB) | $FB = 2.0\,\dfrac{\overline{O_i - M_i}}{\bar{O} + \bar{M}}$ | Hanna and Chang (2012) |
| Normalized mean-square error (NMSE) | $NMSE = \dfrac{\overline{(O_i - M_i)^2}}{\bar{O} \times \bar{M}}$ | Hanna and Chang (2012) |
| Normalized absolute difference (NAD) | $NAD = \dfrac{\overline{|O_i - M_i|}}{\bar{O} + \bar{M}}$ | Hanna and Chang (2012) |


*Data availability*. MUNICH input and output data, and scripts to generate the figures and calculations are available on GitHub (https://github.com/quishqa/MUNICH_VEIN_SP) and Zenodo (http://doi.org/10.5281/zenodo.4168056). MUNICH (v1.0) is available on http://cerea.enpc.fr/munich/index.html and Zenodo (http://doi.org/10.5281/zenodo.4168985). VEIN
can be installed from CRAN, and it is also available in Zenodo (http://doi.org/10.5281/zenodo.3714187). Additional information and help are available by contacting the authors.

*Author contributions*. MGC performed the simulations and prepared the manuscript with the support of all co-authors. MGC, MFA and YZ designed the experiment. SIE provided the emissions and street morphology information. YK provided support
to set up and run MUNICH. MGC, YZ, MFA, and SIE discussed the results.

*Competing interests*. The authors declare that they have no conflicts of interest.

*Acknowledgements*. The authors thank CETESB (São Paulo State Environmental Protection Agency) for providing air
pollution and meteorological data, the support from CAPES (Coordenadoria de Aperfeiçoamento de Pessoal de Nível Superior), CNPq (Conselho Nacional de Desenvolvimento Científico e Tecnológico), and FAPESP (Fundação de Amparo à Pesquisa do Estado de São Paulo, process 2016/18438-0), and the Wellcome Trust (subaward from Yale University to Northeastern University, subcontract number GR108374).

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
