# Peer review of "Simulation of $O_3$ and $NO_X$ in Sao Paulo street urban canyons with VEIN (v0.2.2) and MUNICH (v1.0)"

_Geoscientific Model Development, 2020_

## Referee Comment (RC1) · Anonymous Referee #1 · 22 Nov 2020

The paper assumes that the pollutant concentration is mainly contributed by the local sources, not regional sources. In a lot of cases, just the local emission amount may not be accurate. What about regional emission? In particular, O3 typically is a regional source that can be transported from a far way. Without quantifying the ratio between local and regional sources, it is difficult to evaluate the reliability of the model.

2.2 VEIN emission model Line 140-142: "Therefore, if we consider the mean emission-factor ratio times the mentioned traffic flow ratio results that the NOX emissions should be approximately 2.37 higher." Is the suggested ratio of 2.37 considering contributions from both light vehicles and heavy vehicles?

[Figure]

Line 145: "We choose Wednesday emission as a typical weekday and Saturday emission for the weekend." How much difference between typical Saturday and Sunday traffic in SPMA?

2.3.3 Building height and street width Line 176 "Building height is retrieved from the World Urban Database and Access Portal Tools project (WUDAPT) for SPMA (Fig. 3)." How well is WUDAPT describing building height? Especially, LCZ1, "compact high-rise", is having a description of "height of roughness elements >25m". It is also mentioned in line 226 that "Paulista Avenue domain is more uniform, presenting urban canyons with a mean building height of 45 meters (LCZ1 - Compact high rise).", how is the value of 45 meters obtained? How sensitive is the model to these building height values?

2.3.4 Background concentration Line 195-198 "With that in mind, by using the mean wind field from WRF simulation for the study period, we select Ibirapuera AQS (83 shown in Fig. 4) measurement as background concentration, which, according to the wind field, advect pollutants to Pinheiros station (99) and Cerqueira Cesar (83) as can be seen in Fig. 4."

Is the difference of wind direction from mean during the study period justifying the choice of a single AQS at upwind to provide background concentration. Surely, that single station cannot be upwind for all year round?

Figure 4 Minor: in figure Cerqueira Cesar (red diamond) has number 91 instead of 83 as in line 197 and caption. Typo?

2.5 Model set up Line 215 "VEIN calculates the emissions for the whole SPMA" Line 219-220 "The red lines are the street links used by VEIN to calculate the emissions, and the yellow rectangle the urban canyon selected for comparison against observation." I am not quite sure what this means. Are red lines in figure 5(a), (b) all street links in the domain? If there are street links that are not used by VEIN to calculate the emission? If so, how is their emission calculated?

3.2 Emission adjustment Line 263-264 "We ran different scenarios with increased NOX and VOCs emission from VEIN. The best results were produced when doubled the NOX and VOC emissions. This scenario is called MUNICH-Emiss." If there is any reason picking 2x as the adjusted emission? Would it perform better if higher emission, e.g., 2.5x, is used?

4 Discussion and conclusions Line 396 "calibrated emissions." What does this mean? Is it the MUNICH-Emiss? Or is it calibrated in some way?

---

## Author Comment (AC1) · 9 Jan 2021

Dear reviewer and editor,

Thanks for your accurate observations, time, and dedication in reviewing this manuscript. We covered all your points shown below.

Many thanks

Comment 1

The paper assumes that the pollutant concentration is mainly contributed by the local sources, not regional sources. In a lot of cases, just the local emission amount may

not be accurate. What about regional emission? In particular, O3 typically is a regional source that can be transported from a far way. Without quantifying the ratio between local and regional sources, it is difficult to evaluate the reliability of the model.

Reply: Thank you for this important observation. Previous studies in SPMA identify the vehicular fleet as the main source of air pollution (Andrade et al 2015, 2017). According to Sao Paulo Environmental Agency (CETESB), in 2014 the vehicular fleet was responsible for emitting 97% of CO, 82 % of VOCs, 78 % of NOX, and 40% of particulate matter (PM) emissions in SPMA (CETESB, 2015). To clarify the importance of the local sources, we include the following paragraph in section 2.3.1 Emissions and street link coordinates: "The vehicular fleet is the principal source of air pollution in SPMA (Andrade et al., 2015, 2017). The particularity of this fleet is the extensive use of biofuels (i.e. gasohol, ethanol, and biodiesel). During 2014, vehicular emissions were responsible for emitting 97 % of CO, 82 % of VOCs, 78 % of NOX, and 40 % of particulate matter (CETESB, 2015)."

On the other hand, as we described in section 2.3.4, background concentration in air quality modeling in street canyons accounts for the proportion of air pollutants that aren't emitted in the simulated street-network (Vardoulakis et al., 2003). In our case, we used concentrations of O3, NO2, NO from the Ibirapuera air quality station as background concentration. To explicitly state the air pollutants used as background concentrations, we add the following sentences in section 2.3.4: "In this work, measurements of O3, NO2, and NO in Ibirapuera AQS were used as background concentrations."

**Comment 2**

2.2 VEIN emission model Line 140-142: "Therefore, if we consider the mean emission factor ratio times the mentioned traffic flow ratio results that the NOX emissions should be approximately 2.37 higher." Is the suggested ratio of 2.37 considering contributions from both light vehicles and heavy vehicles?

Reply: Thank you for your comment. The answer is yes. As we detected less traf-
fic flow by comparing GPS with travel demand models' outputs of light and heavyduty vehicles, it should be less emissions. That paragraph was reformulated and we recalculated the ratios between real-world and laboratory emission-factors to produce adjustment factors, already implemented in a newer version of the VEIN model (https://atmoschem.github.io/vein/reference/ef\_cetesb.html). Specifically, the real-world emissions factors for light-duty vehicles and trucks 1.11 and 1.38 times higher than the emission factors reported by the environmental authority (CETESB, 2015).

We rephrase the paragraph as follows: "The emissions dataset presents two aspects that need to be discussed. The first one is that there are some differences between the traffic flow from travel demand model outputs (TDM) and GPS (Ibarra-Espinosa et al., 2019, 2020). The ratio between traffic flows from TDM and GPS for our study area is 2.22. Regarding the emissions factors used to estimate the emissions, they are based on the average measurement of emissions certification tests (CETESB, 2015), therefore, they may underestimate real-drive emissions (Ropkins et al., 2009). For instance, the real-world emission factors derived from tunnel measurements in São Paulo for NOX were 0.3 g km-1 for light vehicles and 9.2 g km-1 for heavy vehicles (Pérez-Martínez et al., 2014), while the respective fleet-weighted CETESB (2015) emission factors are 0.26 g km-1 and 6.68 g km-1, as shown in Fig. S1 in Supplement, resulting in ratios of 1.11 and 1.38. Then, if we consider the mean emission-factor ratio (1.11 + 1.38)/2, times the mentioned traffic flow ratio (2.22) results that the NOX emissions might be approximately 2.73 higher than the estimated using pure CETESB (2015) data. Consequently, we expect that air quality simulations for NOx might be lower than observations."

Comment 3

Line 145: "We choose Wednesday emission as a typical weekday and Saturday emission for the weekend." How much difference between typical Saturday and Sunday traffic in SPMA?
Reply: Thank you for your comment. One of the advantages of VEIN is the use of vehicle GPS data that allows a traffic estimation and therefore a better temporal and spatial emission profile. Figure 1 shows the mean emission from all street links from the Pinheiros neighborhood for NOX and VOCs. In the case of NOx emissions, Sunday total emissions are 25 % lower than Saturday total emissions, while in the case of VOC the values are almost the same. According to Ibarra et al. (2020), the difference between NOX emission during the weekday and the weekend is explained by the Buses contribution, which is lower during the weekend, and even lower during Sunday. Figure 1 is added to Supplement. This is an important point to explain NOX and NO overestimation during Sunday for both Pinheiros and Paulita Avenue urban canyons.

**Comment 4**

2.3.3 Building height and street width Line 176 "Building height is retrieved from the World Urban Database and Access Portal Tools project (WUDAPT) for SPMA (Fig. 3)." How well is WUDAPT describing building height? Especially, LCZ1, "compact high-rise", is having a description of "height of roughness elements >25m". It is also mentioned in line 226 that "Paulista Avenue domain is more uniform, presenting urban canyons with a mean building height of 45 meters (LCZ1 - Compact high rise).", how is the value of 45 meters obtained? How sensitive is the model to these building height values?

Reply: Thank you for noticing this. We explain this point by adding the following paragraph in section 2.2.3: "We retrieve the building height from the updated URB-PARM.TBL file from WRF-Chem simulations in Pellegati et al. (2019). This file was built with the information described in Stewart et al. (2014), and contains the geomorphological and radiative parameters for each WUDAPT LCZ to be used in the Building Environment Parameterization (BEP) simulation in Pellegati et al. (2019)."

We believe that WUDAPT offers a good reference building height value rather than use a constant building height value. Certainly, this information needs to be improved by
comparing it with other data sources as Google Earth or by in-situ measurements. We rephrased line 399 in the Discussion and Conclusions section as the following: "On the other hand, now Google Earth allows new features like 3D view, that together with in-situ measurements, can improve WUDAPT building heights estimates."

Furthermore, we also ran a test with different constant building heights (i.e. 30 m, 50 m, 70 m). MUNICH is coherent with previous results where dispersion is restricted in deep urban canyon leading to higher pollutants concentrations (Afiq et al. 2012). As shown in Fig. 2 higher concentrations of NOX are produced inside the urban canyon when we increase the building height, this leads to a decrease of O3, by its reaction with the NOx. As we can see, background concentration and emission rates have a higher impact than the building height in air quality simulation with MUNICH.

**Comment 5**

2.3.4 Background concentration Line 195-198 "With that in mind, by using the mean wind field from WRF simulation for the study period, we select Ibirapuera AQS (83 shown in Fig. 4) measurement as background concentration, which, according to the wind field, advect pollutants to Pinheiros station (99) and Cerqueira Cesar (83) as can be seen in Fig. 4." Is the difference of wind direction from mean during the study period justifying the choice of a single AQS at upwind to provide background concentration. Surely, that single station cannot be upwind for all year round?

Reply: Thanks for bringing this important question. When we analyzed the wind fields generated by WRF simulations we can see that there is a different behavior during the daylight (Fig 3.a) and nighttime (Fig 3.b). During daylight, there is the advection from Ibirapuera AQS to Pinheiros and Cerqueira Cesar AQS, whereas during night time west winds are predominant. As ozone concentrations during the night are low, it is more important to use information from air quality stations that measure the ozone upwind Pinheiros and Cerqueira Cesar AQS. Still, as noted in the discussion sec-
tion, it could be better to use air quality model results as background concentration for MUNICH, not only for a better background concentration estimate but also to address this wind direction implication. Figure 2 is added to Supplement, and we clarify that this assumption is valid during daylight.

**Comment 6**

Figure 4 Minor: in figure Cerqueira Cesar (red diamond) has number 91 instead of 83 as in line 197 and caption. Typo?

Reply: Thank you for noticing this. Yes, Red diamond should have the number 83. It is now corrected in the manuscript.

**Comment 7**

2.5 Model set up Line 215 "VEIN calculates the emissions for the whole SPMA" Line 219-220 "The red lines are the street links used by VEIN to calculate the emissions, and the yellow rectangle the urban canyon selected for comparison against observation." I am not quite sure what this means. Are red lines in figure 5(a), (b) all street links in the domain? If there are street links that are not used by VEIN to calculate the emission? If so, how is their emission calculated?

Reply: Thanks for bringing this up. As detailed in section 2.3.1, VEIN produces emissions for all the street links in SPMA. This information is a simple feature (sf) class object that contains a column with the Municipality/Neighborhood name of each street link. For this work, we subset the street links for Pinheiros neighborhood, and for the neighborhoods that contain the Paulista Avenue urban canyon. Therefore, the red lines in figure 5(a), (b) in the manuscript are a selection of the original VEIN output for SPMA. We clarify this in section 2.5 by adding the following sentence: "VEIN produces emissions for all the street links in SPMA. This information can be filtered by the neighborhood name of the street links. We subset that information for Pinheiros neighborhood, and for the neighborhoods that contain the Paulista Avenue urban canyon."
**Comment 8**

3.2 Emission adjustment Line 263-264 "We ran different scenarios with increased NOX and VOCs emission from VEIN. The best results were produced when doubled the NOX and VOC emissions. This scenario is called MUNICH-Emiss." If there is any reason picking 2x as the adjusted emission? Would it perform better if higher emission, e.g., 2.5x, is used?

Reply: Thank you for your comment. We performed sensitivity tests with different emissions increment scenarios: the original emissions (original VEIN output), doubled emissions, tripled emissions, and quadrupled emissions. We noticed that the increment of emissions improves ozone simulation. Nevertheless, the increment could lead to unreasonable NOX concentrations, as in the case of the quadrupled emission scenario. The tripled emission scenario presented less error in magnitude than the doubled emission scenario, but it presented a lower Pearson correlation coefficient than the doubled emission scenario for NO, NO2, and NOX. To decide the better scenario, we used the index of agreement statistic (IOA). The doubled emission scenario presented higher IOA values for NO, NO2, and NOX. For that reason, we chose the doubled emission scenario as MUNICH-Emiss. We didn't test for 2.5x as the MUNICH-Emiss scenario already provided good results and reached Hanna and Chang (2012) performance criteria.

**Comment 9**

4 Discussion and conclusions Line 396 "calibrated emissions." What does this mean? Is it the MUNICH-Emiss? Or is it calibrated in some way?

Reply: Yes, in this case, "calibrated emissions" refers to the scenario where emissions are doubled. We have explicitly stated on the manuscript by adding "(i.e. MUNICH-Emiss scenario)" on line 396.

References
Afiq, W. M. Y., Azwadi, C. S. N. and Saqr, K. M.: Effects of buildings aspect ratio, wind speed and wind direction on flow structure and pollutant dispersion in symmetric street canyons: A review, Int. J. Mech. Mater. Eng., 7(2), 158–165, 2012.

Andrade, M. de F., Kumar, P., de Freitas, E. D., Ynoue, R. Y., Martins, J., Martins, L. D., Nogueira, T., Perez-Martinez, P., de Miranda, R. M., Albuquerque, T., Gonçalves, F. L. T., Oyama, B. and Zhang, Y.: Air quality in the megacity of São Paulo: Evolution over the last 30 years and future perspectives, Atmos. Environ., 159, 66–82, doi:10.1016/j.atmosenv.2017.03.051, 2017.

Andrade, M. de F., Ynoue, R. Y., Freitas, E. D., Todesco, E., Vara Vela, A., Ibarra, S., Martins, L. D., Martins, J. A. and Carvalho, V. S. B.: Air quality forecasting system for Southeastern Brazil, Front. Environ. Sci., 3(February), 1–14, doi:10.3389/fenvs.2015.00009, 2015.

CETESB: Qualidade do ar no estado de São Paulo 2014, São Paulo. [online] Available from: http://www.cetesb.sp.gov.br/ar/qualidade-do-ar/31-publicacoes-e-relatorios, 2015.

Hanna, S. and Chang, J.: Acceptance criteria for urban dispersion model evaluation, Meteorol. Atmos. Phys., 116(3–4), 133–146, doi:10.1007/s00703-011-0177-1, 2012.

Ibarra-Espinosa, S., Ynoue, R. Y., Ropkins, K., Zhang, X. and de Freitas, E. D.: High spatial and temporal resolution vehicular emissions in south-east Brazil with traffic data from real-time GPS and travel demand models, Atmos. Environ., 222(May 2019), 117136, doi:10.1016/j.atmosenv.2019.117136, 2020.

Ibarra-Espinosa, S., Ynoue, R., Giannotti, M., Ropkins, K. and de Freitas, E. D.: Generating traffic flow and speed regional model data using internet GPS vehicle records, MethodsX, 6, 2065–2075, doi:10.1016/j.mex.2019.08.018, 2019.

Pérez-Martínez, P. J., Miranda, R. M., Nogueira, T., Guardani, M. L., Fornaro, A., Ynoue, R. and Andrade, M. F.: Emission factors of air pollutants from vehicles mea-
sured inside road tunnels in São Paulo: case study comparison, Int. J. Environ. Sci. Technol., 11(8), 2155–2168, doi:10.1007/s13762-014-0562-7, 2014.

Ropkins, K., Beebe, J., Li, H., Daham, B., Tate, J., Bell, M. and Andrews, G.: Real-World Vehicle Exhaust Emissions Monitoring: Review and Critical Discussion, Crit. Rev. Environ. Sci. Technol., 39(2), 79–152, doi:10.1080/10643380701413377, 2009.

Stewart, I. D., Oke, T. R. and Krayenhoff, E. S.: Evaluation of the "local climate zone" scheme using temperature observations and model simulations, Int. J. Climatol., 34(4), 1062–1080, doi:10.1002/joc.3746, 2014.

Vardoulakis, S., Fisher, B. E. A., Pericleous, K. and Gonzalez-Flesca, N.: Modelling air quality in street canyons: A review, Atmos. Environ., 37(2), 155–182, doi:10.1016/S1352-2310(02)00857-9, 2003.
**Fig. 1.** Mean emission from all street links from the Pinheiros neighborhood for (a) NOX and (b) VOCs for a typical week.

---

## Referee Comment (RC2) · Anonymous Referee #2 · 9 Feb 2021

This manuscript demonstrates the street scale air quality modelling system and its evaluation for the city of Sao Paulo. The authors present it as the operational forecast system. However, the forecast system implies that the future atmospheric pollution can be predicted. And "forecast system" seems to be an improbable description of it (Line 85), given that you used real-time air quality observations to force your air pollution forecast. The current system is rather suitable for policymaking and future urban planning or post-accident analysis.

The meteorological driver (WRF) evaluation was performed in a slightly opaque manner since the authors did not mention neither the location (and number) of meteorological

observation sites against which the model was evaluated nor the period of evaluation (perhaps of the same time extent as MUNICH runs). It is also unclear if the WRF output from D03 domain only was evaluated.

Perhaps, the authors could try to pinpoint the cause of large NOx and NO underestimation at Pinheiros AQS during Oct 8-9. Could it be associated with local meteorological conditions (probably unaccounted effect of nearby river, inversion etc.) or very local emissions just during those 2 days?

The reasons behind two distinct peaks in NOx and NO observations (not captured by MUNICH) at both AQSs during night time seem to be ambiguous. Did the authors check if those are associated with meteorology? In case they are not related to any issue with meteorology, why did not the authors adjust emissions (one vs. two peaks) to fit the observed concentrations during the nights?

Overall, this study is very interesting. However, the manuscript requires additional clarifications and corrections listed below in the specific comments and technical corrections.

Specific comments:

Line 125: "street links" is confusing definition of roads, in particular for those who have never dealt with VIEN model. Perhaps, you should define it before using.

Lines 127-128: Could you please elaborate a bit on how the vehicular composition was obtained from GPS dataset and CETESB (2015) report? The report appears to be in Portuguese language and it might be hard to understand for those who speak/read English only.

Line 140: The only number which fits the early-mentioned emission factors is 1.46. What is the 0.68 about?

Lines 183-185: "The number of lanes is provided by the OpenStreetMap dataset. . ." and "Most OpenStreetMap streets do not include the number of lanes for this region. . ."

seem to contradict each other. Both sentences should be reformulated to fit the method you actually used in the manuscript.

Lines 196-197: The Ibirapuera AQS (83) does not seem to be the optimal location for background concentration if you look at the mean wind field of upstream region. Perhaps, the mean of observed concentrations from (83) and (94) AQSs would fit better for MUNICH's forcing. Did the authors consider/try such forcing?

Line 276: phrase "MUNICH uses the same emission profile for the weekend and weekdays" is in contradiction with the section 2.3.1 and Figure 1, where emissions for weekdays and weekends are claimed to be different.

Table 4: There are often exceptions, but the fact that the correlation values equal strictly 1 in all 3 cases for ozone is unfortunately hard to believe. Maybe you rounded values or made some error during computations. Adding an extra digit for R values would be a good idea. Since the "Background" concentrations are also observed, it is unclear why authors evaluated and compared them with the street observations and what they tried to achieve by doing that (quality control?).

Line 332: "in MUNICH NOx and NO peak happening before observation." Since you have many models and databases interfaced with each other, such mismatch in simulated concentrations could have happened because you did not match timings of datasets and models having them all, for example, in UTC. Are you sure the models and data were perfectly matched?

Technical corrections:

Line 95: "before of no precipitation in" probably change to "before dry weather conditions in"

Line 136: please add reference for TDM

Lines 146-149: The unit of flux [ug / km / h] is confusing (in Figure 1). Shouldn't it be something like [ug / km*2 / h], typo?

Line 161/ Figure 2: "WRF simulation domains for domains of..." please rephrase

Line 196: Cerqueira Cesar (83), should not that be 91 (similar typo in Figure 4)?

Line 220: "rectangle the urban canyon" change to "rectangle is the urban canyon"

Line 229: "adn Paulista Avenue" change to "and Paulista Avenue"

Line 309: "We also perform additional" change to "We also performed an additional"

Line 319: "COV-limited regime" isn't it "VOC-limited regime"?

Line 320: "with lead to" what does that mean, typo?

Line 331: "but still higher than 0.5" it is imprecise as there are R values of 0.4 and 0.2 in the Table 5.

Lines 341, 345: "Note that no O3 observation for Paulista Avenue." seems grammatically incorrect sentence.

Line 386: "As the main source of superficial NO" probably you should write "... of elevated NO"

---

## Author Comment (AC2) · 15 Mar 2021

Dear reviewer and editor,

Thanks for your important observations, time, and dedication in reviewing this manuscript. We covered all your points as shown below.

Many thanks

Comment 1

This manuscript demonstrates the street scale air quality modelling system and its

evaluation for the city of Sao Paulo. The authors present it as the operational forecast system. However, the forecast system implies that the future atmospheric pollution can be predicted. And "forecast system" seems to be an improbable description of it (Line 85), given that you used real-time air quality observations to force your air pollution forecast. The current system is rather suitable for policymaking and future urban planning or post-accident analysis.

Reply: Thank you for pointing this out. Indeed, the forecast system will be achieved using a photochemical grid model to provide background concentration to MUNICH (like the case of SinG model described in Kim et al. (2018)) or an air quality on-line model that can provide both meteorological information and background concentrations. We briefly mention this point in the Discussion and Conclusions section when we detailed that output from photochemical grid models can improve MUNICH background concentration. Following your observation, we changed "forecast system" to "street-level air quality modeling system". The new paragraph is as follow:

"As the management of secondary pollutants remains a challenge in SPMA, we aim to evaluate MUNICH operational street-network model to simulate O3 and NOx concentration inside urban canyons, coupled with the VEIN emission model, to build a street-level air quality modeling system. This modeling system can be used in air quality and traffic management of Sao Paulo neighborhood, in studies of health effects from traffic emission exposure, in future urban planning, and post-accident analysis."

**Comment 2**

The meteorological driver (WRF) evaluation was performed in a slightly opaque manner since the authors did not mention neither the location (and number) of meteorological observation sites against which the model was evaluated nor the period of evaluation (perhaps of the same time extent as MUNICH runs). It is also unclear if the WRF output from D03 domain only was evaluated.

Reply: This is an important point. We performed the model evaluation only for our
study period, the week from October 6th to 13th, 2014 as described in Table 2. We only evaluated the output from the finest domain (D03) as this is the domain that provided meteorological information to MUNICH. Figure 4 shows the air quality station locations, but not all the stations have meteorological information. Some air quality stations (AQS) only measure pollutant concentrations together with some meteorological parameters. During this period a total of 16 AQS have meteorological data. Only eight AQS measured temperature (T2), relative humidity (RH2), wind speed (WS), and wind direction (WD); five AQS measured only wind speed and direction; and three AQS measured only temperature and relative humidity. We updated Figure 4 to point the AQS with meteorological information. We also clarify these points by the following paragraph in section 2.3.2 WRF simulation:

"Before using the WRF simulation outputs for MUNICH modeling, a model verification is performed. Model verification was carried out for the same period as MUNICH runs and for the finest domain output (D03). We used meteorological information from 16 air quality stations which locations are shown in Figure 4."

**Comment 3**

Perhaps, the authors could try to pinpoint the cause of large NOx and NO underestimation at Pinheiros AQS during Oct 8-9. Could it be associated with local meteorological conditions (probably unaccounted effect of nearby river, inversion etc.) or very local emissions just during those 2 days?

Reply: Thanks for bringing this up. The underestimation during Oct-8-9 can be explained by a very local emission episode as it did not happen in the Paulista Avenue domain, at least during October 9th where data is available. Still, underestimation of NOX concentration is caused by underestimation of NO concentration which is produced by a lower background concentration and an underestimation of emission factors as discussed in Section 2.3.1 Emissions and street links coordinates. Another factor is that MUNICH uses a single-day emission profile to represent weekdays emission,
which can not account for the daily emission variation during the week. Meteorological factors as the overestimation of the wind speed by WRF model enhances dispersion. We add this information in section 3.2. Emission adjustment by rephrasing the paragraph as follows: "NOx and NO simulations are still underpredicted, but NO2 is in the same magnitude as observations. NOx underprediction is still mainly attributed to the underprediction of NO, especially during October 8th, 9th, and 10th where high observational values of NO were recorded. NO underestimation is explained by the lower NO background concentration, the underestimation of emissions, and the use of a single-day emission profile to represent all weekdays. Wind speed overestimation also affects this underestimation as it enhances dispersion. However, MUNICH can better represent the observed high concentration during Saturday 11th, as MUNICH uses the same emission profile for the weekend and weekdays, this high simulated NO concentration resulted from the influence of meteorology. "

**Comment 4**

The reasons behind two distinct peaks in NOx and NO observations (not captured by MUNICH) at both AQSs during night time seem to be ambiguous. Did the authors check if those are associated with meteorology? In case they are not related to any issue with meteorology, why did not the authors adjust emissions (one vs. two peaks) to fit the observed concentrations during the nights?

Reply: Thanks for this observation. Errors during nighttime can be caused by wrong representations of meteorology by WRF and by errors in the emission profile. In the case of meteorology, it is common that WRF presents troubles to represent the planetary boundary layer height during nighttime (Hu et al., 2012; McNider & Pour-Biazar, 2020). On the other hand, as shown in the emission profile during weekday and weekend days in Figure 1, NOX emissions do present two emission peaks during 7 hours and 16 hours, and a smaller emission peak around 23 hours, it is probable that this nighttime peak was underestimated. We add the following text in Section 3.3. Application for the Paulista Avenue:
"As in Pinheiros domain, MUNICH did not capture the two peaks of NO and NOX during nighttime. This is caused by WRF limitation in representing planetary boundary layer height during nighttime (Hu et al., 2012; McNider & Pour-Biazar, 2020). Also as shown in Fig. 1a, NOX emission profile during weekday present two peaks during daylight at 7 hours and 16 hours (Local Time), and a smaller emission peak around 23 hours, it is probable that this nighttime peak was underestimated."

**Comment 5**

Line 125: "street links" is confusing definition of roads, in particular for those who have never dealt with VIEN model. Perhaps, you should define it before using.

Reply: Agreed. Street links are segments of roads split at each vertex. Then, a road is composed of many links. We added this definition in section 2.3.1 Emissions and street links coordinates.

**Comment 6**

Lines 127-128: Could you please elaborate a bit on how the vehicular composition was obtained from GPS dataset and CETESB (2015) report? The report appears to be in Portuguese language and it might be hard to understand for those who speak/read English only.

Reply: The details about transforming GPS data into vehicular flow are described in Ibarra-Espinosa et al (2019). The details about using these GPS traffic flow to estimate vehicular emissions are described by Ibarra-Espinosa et al (2020). The CETESB report in Portuguese is cited only to cite the source of the emissions factors. CETESB measures and receives emissions laboratory measurements and report the emission factors. The references are below in this reply.

**Comment 7**

Line 140: The only number which fits the early-mentioned emission factors is 1.46. What is the 0.68 about?
Reply: We detected that real-world heavy trucks emissions factors from tunnel measurements (9.2 g km-1) are higher than laboratory measurements (6.3 g km-1) resulting in a ratio of 9.2/6.68 = 1.38. In the case of light vehicles, tunnel measurements emission factors (0.3 g km-1) are lower than laboratory measurements (0.44 g km-1), resulting in a ratio of 0.3/0.44 = 0.68. Recalling that the traffic is underestimated 2.2 times, the average of ratio emission factors (0.68+1.37)/2 times 2.2, results in approx in 2.3. This was confusing in the text and we apologize for that. But then, we realized that, as the tunnel emission factors are representative of the circulating fleet, we should weigh the CETESB emission factors by the circulating fleet as well. Then, we re-wrote the whole paragraph to improve the clarity as mentioned here:

"The emissions dataset presents two aspects that need to be discussed. The first one is that there are some differences between the traffic flow from travel demand model outputs (TDM) and GPS (Ibarra-Espinosa et al., 2019, 2020). The ratio between traffic flows from TDM and GPS for our study area is 2.22. Regarding the emissions factors used to estimate the emissions, they are based on the average measurement of emissions certification tests (CETESB, 2015), therefore, they may underestimate real-drive emissions (Ropkins et al., 2009). For instance, the real-world emission factors derived from tunnel measurements in São Paulo for NOX were 0.3 g km-1 for light vehicles and 9.2 g km-1for heavy vehicles (Pérez-Martínez et al., 2014), while the respective fleet-weighted CETESB (2015) emission factors are 0.26 g km-1 and 6.68 g km-1, as shown in Fig. S1 in Supplement, resulting in ratios of 1.11 and 1.38. Then, if we consider the mean emission-factor ratio (1.11 + 1.38)/2, times the mentioned traffic flow ratio (2.22) results that the NOX emissions might be approximately 2.73 higher than the estimated using pure CETESB (2015) data. Consequently, we expect that air quality simulations for NOX might be lower than observations."

**Comment 8**

Lines 183-185: "The number of lanes is provided by the OpenStreetMap dataset. . ." and "Most OpenStreetMap streets do not include the number of lanes for this region.
. ." seem to contradict each other. Both sentences should be reformulated to fit the method you actually used in the manuscript.

Reply: Agreed. The paragraph is rephrased as: "Most OpenStreetMap streets do not include the number of lanes for this region, therefore, they are hole-filled with the average by type of street. Then, street link width is calculated by assuming 3 m of line width and by adding 1.9 m to each side of the street as sidewalk width."

**Comment 9**

Lines 196-197: The Ibirapuera AQS (83) does not seem to be the optimal location for background concentration if you look at the mean wind field of upstream region. Perhaps, the mean of observed concentrations from (83) and (94) AQSs would fit better for MUNICH's forcing. Did the authors consider/try such forcing?

Reply: We chose Ibirapuera because it is located inside a park inside Sao Paulo city. Unfortunately, the air quality station with code 94 (Located at Sao Paulo downtown) does not have measurements of O3, NO, and NO2 for October 2014. So we couldn't consider it as background.

**Comment 10**

Line 276: phrase "MUNICH uses the same emission profile for the weekend and weekdays" is in contradiction with the section 2.3.1 and Figure 1, where emissions for weekdays and weekends are claimed to be different.

Reply: Agreed. Sentence is rephrased as: "However, MUNICH can better represent the observed high concentration during Saturday 11 th. As MUNICH uses the same emission profile for the weekdays and another emission profile for weekends, this high simulated NO concentration resulted from the influence of meteorology."

**Comment 11**

Table 4: There are often exceptions, but the fact that the correlation values equal strictly
1 in all 3 cases for ozone is unfortunately hard to believe. Maybe you rounded values or made some error during computations. Adding an extra digit for R values would be a good idea. Since the "Background" concentrations are also observed, it is unclear why authors evaluated and compared them with the street observations and what they tried to achieve by doing that (quality control?).

Reply: Thanks for this important observation. We added two digits for R values in Table 4, R between observations and background concentration was 0.9785, R between observations and MUNICH scenario was 0.9810, and R between observations and MUNICH-Emiss scenario (doubled emission scenario) was 0.9796. We rounded to two digits to R values to save space in Table 4.

We chose to evaluate background concentration against observation to see the difference between observation and background concentration and mainly to assess the influence of the background concentration in MUNICH simulations as previously shown in Wu et al. (2020).

**Comment 12**

Line 332: "in MUNICH NOx and NO peak happening before observation." Since you have many models and databases interfaced with each other, such mismatch in simulated concentrations could have happened because you did not match timings of datasets and models having them all, for example, in UTC. Are you sure the models and data were perfectly matched?

Reply: We took extremely careful consideration in the input time zone and its transformation to local time for a better visualization of model results. In this sense, all MUNICH input/output (i.e. WRF output, VEIN emissions, and background concentration) are in UTC. Change to local time (America/Sao Paulo) was performed using R functionalities - not manually- to avoid errors.

Response to technical corrections:
1. Line 95: "before of no precipitation in" probably change to "before dry weather conditions in"

Reply: Agreed. Sentence changed to "This period is chosen before dry weather conditions in SPMA"

2. Line 136: please add reference for TDM Lines 146-149: The unit of flux [ug / km / h] is confusing (in Figure 1). Shouldn't it be something like [ug / km\*2 / h], typo?

Reply: Agreed. We added the reference for the TDM (Ibarra-Espinosa et al., 2019, 2020). We chose to plot emissions in ug/km/h because it is the unit that street emissions from VEIN required to be transformed to be read by MUNICH. We updated Figure 1 with emission in g/h which are the units used in VEIN. We also realized that Figure 1 was actually on UTC, now is change to Local Time.

3. Line 161/ Figure 2: "WRF simulation domains for domains of. . ." please rephrase

Reply: Agreed. Sentence changed to "WRF simulation domains of 25 km (D01), of 9 km (D02), and of 1 km (D03) spatial resolution".

4. Line 196: Cerqueira Cesar (83), should not that be 91 (similar typo in Figure 4)?

Reply: Agreed. Corrected to "the red circle shows Cerqueira Cesar AQS (91)."

5. Line 220: "rectangle the urban canyon" change to "rectangle is the urban canyon"

Reply: Agreed and change.

6. Line 229: "adn Paulista Avenue" change to "and Paulista Avenue"

Reply: Agreed and change.

7. Line 309: "We also perform additional" change to "We also performed an additional" Reply: Agreed and change.

8. Line 319: "COV-limited regime" isn't it "VOC-limited regime"?

GMDD
Reply: Agreed and change.

9. Line 320: "with lead to" what does that mean, typo?

Reply: Thank you for noticing this. Sentence corrected to "the increment of NOX emission will lead to a reduction of O3 concentration"

10. Line 331: "but still higher than 0.5" it is imprecise as there are R values of 0.4 and 0.2 in the Table 5.

Reply: Agreed. Rephrased to :

"In this case, R values are lower than those in the Pinheiros case but still higher than 0.4 for NO2 and NOX, confirming that there is a mismatch of simulated concentrations, which is clearer in MUNICH NOX and NO peak happening before observation."

11. Lines 341, 345: "Note that no O3 observation for Paulista Avenue." seems grammatically incorrect sentence.

Reply: Agreed. Change to "Note that O3 observations were not available for Paulista Avenue domain."

12. Line 386: "As the main source of superficial NO" probably you should write ". . . of elevated NO"  $\,$

Reply: Agreed. That was actually a typo, the corrected sentence is "As the main source of surface NO and NO2 emissions in Sao Paulo are vehicles,"

References

CETESB: Emissões veiculares no estado de São Paulo 2014, São Paulo. [online] Available from: https://cetesb.sp.gov.br/veicular/relatorios-e-publicacoes/, 2015.

Hu, X. M., Doughty, D. C., Sanchez, K. J., Joseph, E. and Fuentes, J. D.: Ozone variability in the atmospheric boundary layer in Maryland and its implications for vertical transport model, Atmos. Environ., 46, 354–364, doi:10.1016/j.atmosenv.2011.09.054,
2012.

Ibarra-Espinosa, S., Ynoue, R., Giannotti, M., Ropkins, K. and de Freitas, E. D.: Generating traffic flow and speed regional model data using internet GPS vehicle records, MethodsX, 6, 2065–2075, doi:10.1016/j.mex.2019.08.018, 2019.

Ibarra-Espinosa, S., Ynoue, R. Y., Ropkins, K., Zhang, X. and de Freitas, E. D.: High spatial and temporal resolution vehicular emissions in south-east Brazil with traffic data from real-time GPS and travel demand models, Atmos. Environ., 222(May 2019), 117136, doi:10.1016/j.atmosenv.2019.117136, 2020.

McNider, R. T. and Pour-Biazar, A.: Meteorological modeling relevant to mesoscale and regional air quality applications: a review, J. Air Waste Manag. Assoc., 70(1), 2–43, doi:10.1080/10962247.2019.1694602, 2020.

Pérez-Martínez, P. J., Miranda, R. M., Nogueira, T., Guardani, M. L., Fornaro, A., Ynoue, R. and Andrade, M. F.: Emission factors of air pollutants from vehicles measured inside road tunnels in São Paulo: case study comparison, Int. J. Environ. Sci. Technol., 11(8), 2155–2168, doi:10.1007/s13762-014-0562-7, 2014.

Ropkins, K., Beebe, J., Li, H., Daham, B., Tate, J., Bell, M. and Andrews, G.: Real-World Vehicle Exhaust Emissions Monitoring: Review and Critical Discussion, Crit. Rev. Environ. Sci. Technol., 39(2), 79–152, doi:10.1080/10643380701413377, 2009.

Wu, L., Chang, M., Wang, X., Hang, J. and Zhang, J.: Development of a Real-time On-road emission (ROE v1.0)model for street-scale air quality modeling based on dynamic traffic big data, Geosci. Model Dev., (13), 23–40, doi:gmd-13-23-2020, 2020.

Please also note the supplement to this comment: https://gmd.copernicus.org/preprints/gmd-2020-282/gmd-2020-282-AC2supplement.pdf
2020.

---

## Author Response (AR2)

Dr. Havala Pye
Topical Editor
GMD

Concerning: Accepted manuscript gmd-2020-282 entitled "Simulation of O3 and NOX in Sao Paulo street urban canyons with VEIN (v0.2.2) and MUNICH (v1.0)" by Gavidia-Calderón M. E. et al.

Dear editor,

We are grateful for your help and time during this process. In this last submission, we added the full first name of all the authors and updated the copyright in Figure 5 to "© Google Maps 2019", according to the remarks from the preceding review file validation.
Finally, we added "Department of Atmospheric Sciences" to affiliation 1, to complete the full institutional address.
The rest of the text, tables, and figures remain the same as the accepted version of the manuscript.

Many thanks.

Mario E. Gavidia-Calderón